# The contribution of corporate initiatives to global renewable electricity deployment

Florian Egli [1,2] ✉, Rui Zhang[1], Victor Hopo[1], Tobias Schmidt [1,3] ✉ & Bjarne Steffen [3,4] ✉

Climate change is gaining importance on the agenda of senior decision makers in the private sector. Hence, corporate renewable electricity (RE) procurement may become more relevant to the energy transition. RE100 is the largest corporate initiative to foster RE procurement with 315 corporate members as of 2021. Yet, the contribution of such initiatives to the energy transition remains unclear, because public reporting is aggregated on the global level. Here, we develop an approach to map the electricity procured by RE100 companies to jurisdictions worldwide, which allows estimating whether and where RE100 can have a transformative impact. We find that these companies source electricity in 129 jurisdictions, accounting for <1% of total electricity generation (RE and non-RE), thus dampening the hopes about the impact of RE100 on the global energy transition. RE100 companies procure 1.4% of available RE, exceeding 20% in nine jurisdictions. To increase its impact, RE100 should focus on interim targets and expansion. By 2030, stringent and frequent interim targets could lead to a cumulated additional 361 TWh of RE procured by RE100 companies, and a realistic membership expansion could lead to procurement of 7.7% of globally available RE by RE100 companies.

Fast and deep emission cuts are required to reach the climate targets of the Paris Agreement[1]. This requires a shift from fossil to low-carbon energy carriers[2]. While many governments have pledged ambitious climate action in the context of the 26th United Nations Climate Change Conference in Glasgow (COP26) in 2021, the new commitments do not bring the world on a 1.5 °C pathway[3] and neither do existing ones as submitted in the second generation of nationally determined contributions (NDCs) in the implementation process of the Paris Agreement[4]. Most of these pledges concern the long-term future, for example, net-zero commitments by 2050, but do not result in the required immediate action[5,6]. Against this backdrop, non-state actors—including companies united in corporate initiatives—step up and take actions, and many companies have committed to reaching net-zero emissions by 2050[6–9]. Research to quantitatively analyse corporate climate action is only beginning[8], and it focuses on carbon-intensive companies' efforts[10]. However, many climate pledges come

from less carbon-intensive companies, for example in services, such as Alphabet/Google, Amazon, or Swiss Re. While their contribution to direct emissions is moderate, these actors can—in theory—be important change agents along their supply chains. However, the extent to which these corporate actions actually contribute to the energy transition and thus climate change mitigation (i.e. deliver additional impact over government efforts) remains debatable[11].

Many corporate pledges focus on procuring low-carbon electricity. These Scope 2 emissions often dominate corporate carbon footprints, particularly in the service industry[12]. At the same time, mature renewable energy (RE) technologies, such as solar photovoltaics and wind power, are cost-competitive low-carbon options[13,14]. Consequently, corporate interest in RE has increased in recent years, and corporate initiatives with related goals have grown in number[15]. In 2021, the largest and most prominent corporate initiative to foster RE procurement, the RE100 initiative, had 315 corporate members. These

[1]Energy and Technology Policy Group, ETH Zurich, Zurich, Switzerland. [2]Institute for Innovation and Public Purpose, UCL, London, UK. [3]Institute for Science, Technology and Policy, ETH Zurich, Zurich, Switzerland. [4]Climate Finance Policy Group, ETH Zurich, Zurich, Switzerland. ✉e-mail: florian.egli@gess.ethz.ch; tobiasschmidt@ethz.ch; bjarne.steffen@gess.ethz.ch

firms have committed to procuring 100% of electricity from renewable sources by 2050. In 2021, they jointly procured 340 TWh of electricity, of which 152 TWh or 45% was reported as RE[16]—roughly equalling the total electricity generation of Norway. We use the term "total electricity" (TE) to refer to the total RE and non-RE generation of a country or the total RE and non-RE demand of a company.

Aggregate global numbers may conceal the real effects of RE100 on RE deployment at the country level. For example, corporate RE procured in countries whose electricity generation is dominated by RE (often hydropower), such as Norway, Brazil, or Canada, may provide little added value to the energy transition. Conversely, a high corporate RE demand in countries that rely mostly on coal and natural gas for electricity generation, such as Australia or South Africa, may provide a critical impetus to build RE plants and develop a local RE innovation system[17,18]. Not only may corporate-induced demand lead to RE installations additional to planned RE deployment in such countries, but the associated shifts in political interests may also help overcome strong institutional carbon lock-ins[19], which inhibit ambitious climate policy. Thus, the potential contribution of corporate RE procurement must be analysed at the country level. This is typically challenging because companies consider data on country-level electricity demand confidential. As a result, the impact of corporate RE initiatives on the energy transition remains unclear.

Here, we tackle the challenge of estimating country-level impacts on the energy transition by focusing on the RE100 initiative to disaggregate the electricity demand of its member companies. We split up aggregated reporting of electricity demand by RE100 companies using various proxies for the regional distribution of corporate activities to construct a dataset to project scenarios for near- and mid-term country-level RE deployment (see Methods). Our data reveals that RE100 companies procured 227 TWh of electricity in 129 jurisdictions in 2018, amounting to a negligible share of total (RE and non-RE) electricity generation in these jurisdictions (0.9%, see Supplementary Table 1). As these jurisdictions represent 96% of global electricity generation and 97% of global RE generation, we sometimes refer to these findings as "global". This share is higher in only a few countries, mainly concentrated in Europe,

where RE100 companies account for 3–5% of the total domestic electricity generation. Hence, even if current RE100 companies fully switched their electricity demand to RE, the global effect on electricity systems would be limited.

## Results

### Electricity demand of RE100 firms

Figure 1 provides an overview of RE100 member companies, their progress towards the 100% RE target and the geographic extension of their operations. Data on 185 companies, 88% of RE100 members, was available as of December 2019, with a TE demand of 227 TWh in 2018. On average, RE100 companies were halfway through in decarbonising their electricity demand (52%) in 2018, with a wide variation from 0% to 100% (see Fig. 1a). In total, these companies procured 92 TWh RE in 2018, corresponding to 41% of their electricity demand. The RE share is independent of electricity demand. Progress towards the 100% RE target varies substantially and is unrelated to size. For example, large companies, such as Walmart or Alphabet/Google differ massively with 9% and 100% RE share respectively. On average, these 185 companies have operations in 22 countries (see Methods), with Nestlé and Unilever spanning the largest number of countries (N = 79), as shown in Fig. 1b. Thirty companies are active in a single country (e.g., Gatwick Airport). Most companies are active in the services sector, followed by manufacturing sector (see Fig. 1c).

Overall, the TE demand by RE100 companies is marginal. On the aggregate, it accounts for only 0.9% of the electricity produced in the countries in which these companies have operations (see Supplementary Table 1). However, this grand total—typically the number reported in the academic literature[15,20,21] and industry progress reports[16]—may hide variations in local impacts. By estimating RE100 companies' electricity demand for individual countries (see Methods), we can discover such local variance and identify countries in which the electricity procurement of RE100 companies may cause a substantial increase in RE demand, thus contributing to accelerating the energy transition. Figure 2a displays the share of RE100 companies in the electricity demand in 129 countries, showing that the TE demand from RE100 companies accounts for 0–5% of total domestic electricity generation. The global overview in Fig. 2a shows three geographical

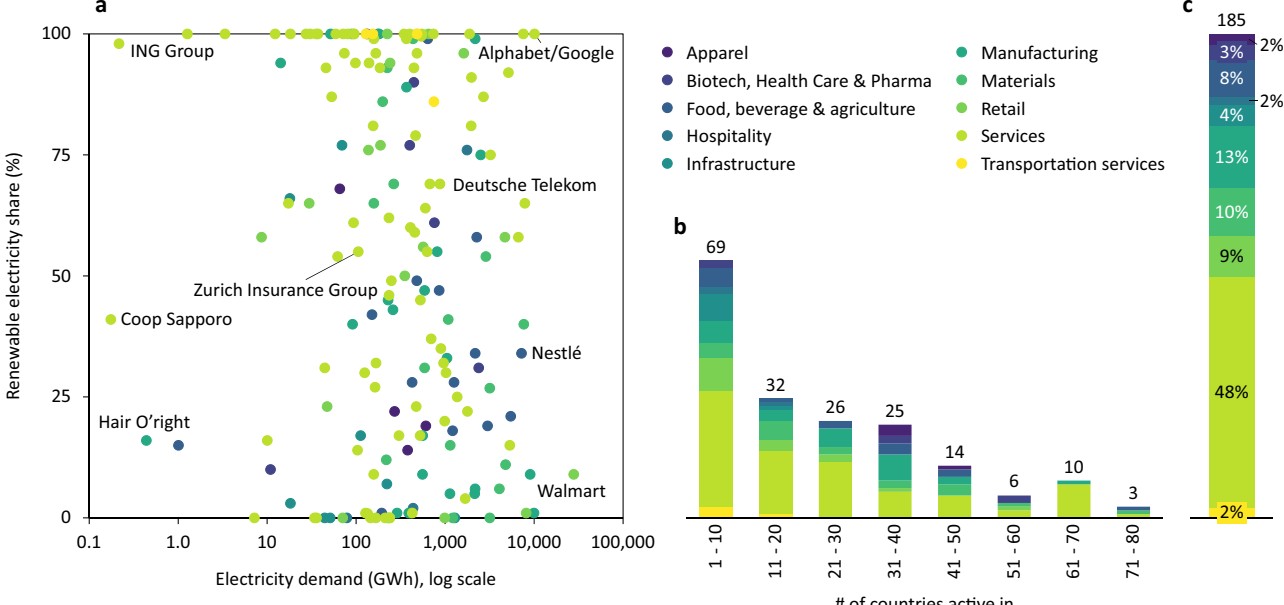

**Fig. 1 | Data overview. a** Total electricity demand by RE100 companies (*x*-axis) and RE share of this demand (*y*-axis) in 2018 (*N* = 185). Some companies are labelled for illustrative purposes. **b** Binned histogram showing the number of countries in which each RE100 member company has operations. For example, three RE100 member companies are active in 71–80 different countries. **c** RE100 companies by sector (see Supplementary Table 4 for sector definitions).

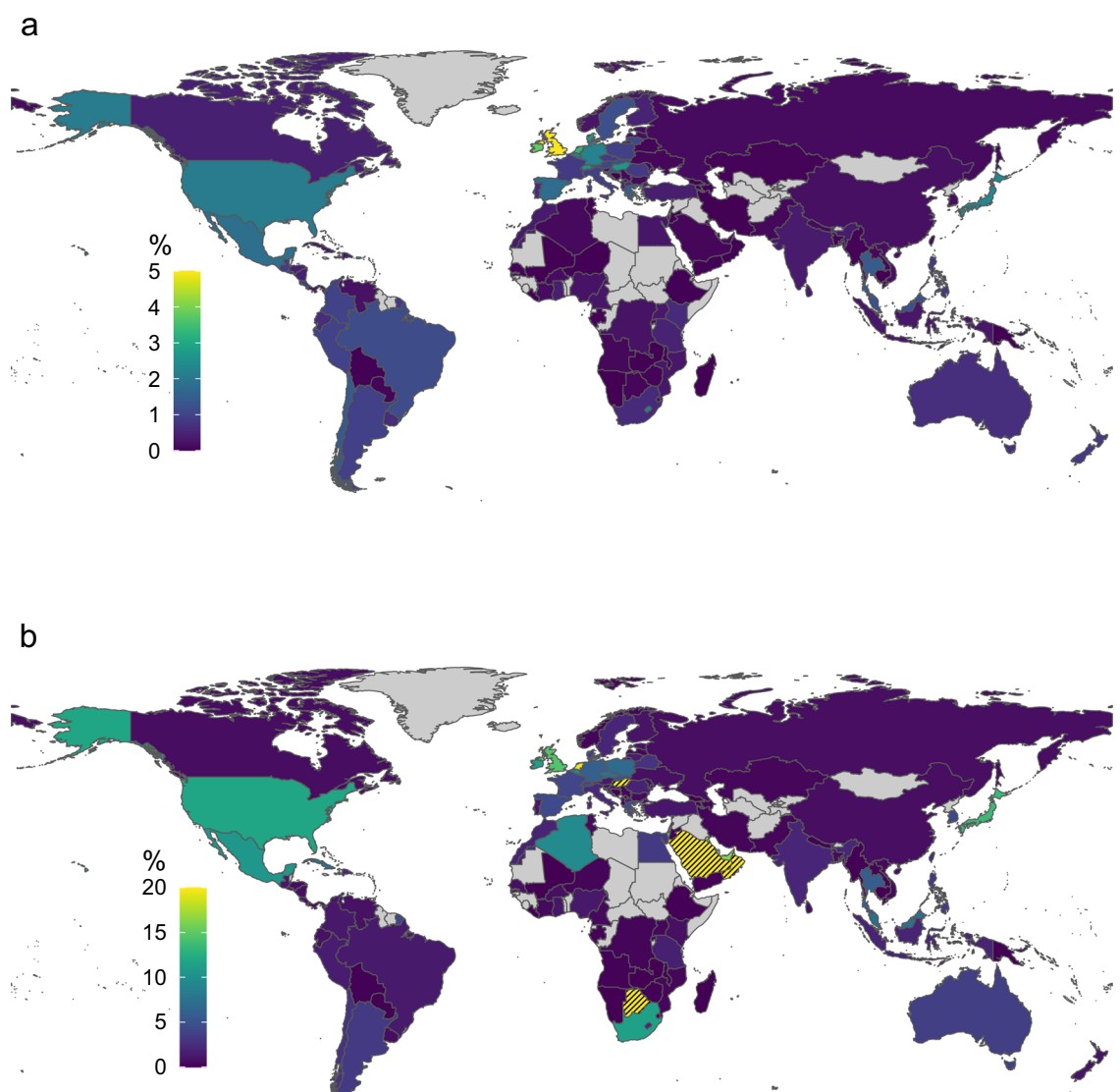

**Fig. 2 | Country-level RE100 impact. a** Total electricity demand by RE100 member companies as a percentage of total domestically generated electricity in 2018. **b** Total electricity demand by RE100 member companies as a percentage of total domestically generated RE in 2018. Countries without the presence of an RE100 company and/or a population of less than 1 million are excluded and shown in grey.

In (**b**), the axis is cut at 20% for readability; hatches denote the nine countries with shares above 20%. From highest to lowest share: Hong Kong, Puerto Rico, Saudi Arabia, Bahrain, Singapore, Trinidad and Tobago, Oman, Botswana, Hungary (see Supplementary Table 2 for shares).

pockets in which these shares are high: Europe, North America and Latin America. At 5%, the United Kingdom tops the list, ahead of Ireland (3.8%) and the Netherlands (3.4%).

Figure 2b shows an alternative way of looking at countries where the impact of RE100 may be particularly large: It displays the TE demand by RE100 in percent of domestic RE generation (3.5% globally, see Supplementary Table 1). The higher this share, the more difficult it may become for RE100 companies to achieve their target by procuring RE domestically. Countries in which the share exceeds 10% are geographically less concentrated and include Japan, South Africa or Mexico. Supplementary Table 2 shows the 20 countries for which the TE demand of RE100 exceeds 10% of currently available RE generation. In Hong Kong, Puerto Rico and Saudi Arabia, this share exceeds 100%, meaning that even the total currently available RE generation would not cover the demand of RE100 companies fulfilling their target. Among these 20 countries, we find a remarkable number of oil-exporting countries, including Saudi Arabia, Bahrain, Trinidad and Tobago, Oman, the United Arab Emirates, Qatar and Kuwait. For these

countries, the impetus of corporations to accelerate the transition to RE may be particularly important because of their strong vested interests standing in the way of decarbonisation[22,23].

In 13 countries, RE100 companies account for more than 2% of total domestic electricity generation, and 67% of the total RE100 electricity demand falls into these countries. Figure 3 shows these countries (3a), together with the TE demand by RE100 companies (3b) and the three largest companies contributing to this demand (3c). We observed that the largest demand from RE100 companies occurs in the United States (roughly 88 TWh), followed by Japan (24 TWh), the United Kingdom (16 TWh) and Germany (13 TWh). China (12.8 TWh), Mexico (5.8 TWh) and India (4.9 TWh) are also among the 10 countries with the largest electricity demand by RE100 companies, but they are not shown in Fig. 3 because this demand still accounts for less than 2% of the total domestic electricity generation. The RE100 top three companies shown in Fig. 3c span a number of different industries (e.g., IT, retail, telecommunications) and Alphabet/Google, Deutsche Telekom, Equinix, Johnson & Johnson, Tesco and Walmart appear in the top

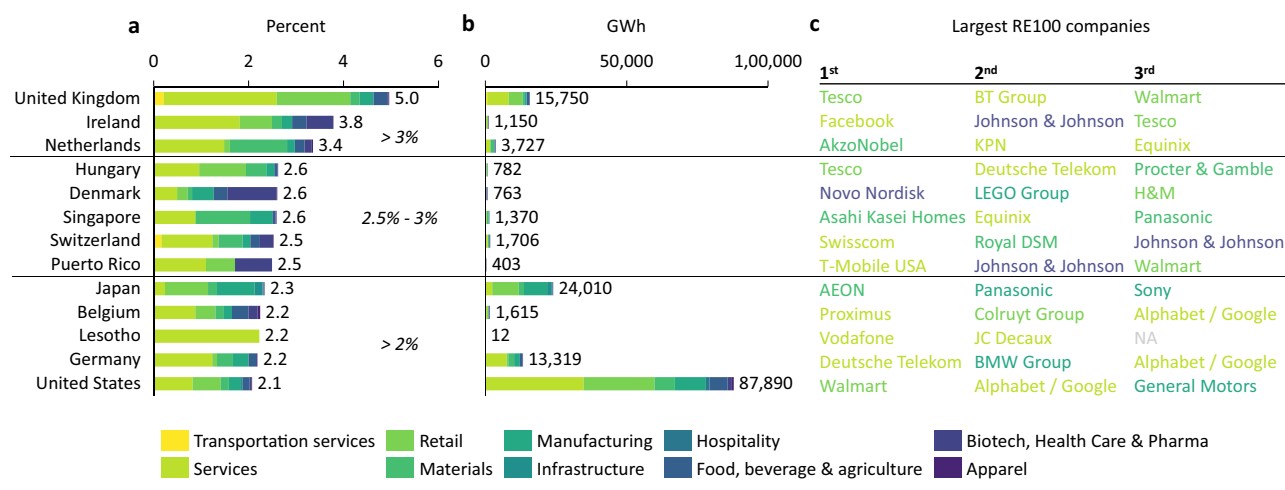

**Fig. 3 | RE100 electricity demand in key markets. a** 13 countries where RE100 electricity demand accounts for more than 2% of domestic electricity generation. **b** Total electricity demand by RE100 companies in GWh. **c** Top three RE100 companies with the largest electricity demand for each country.

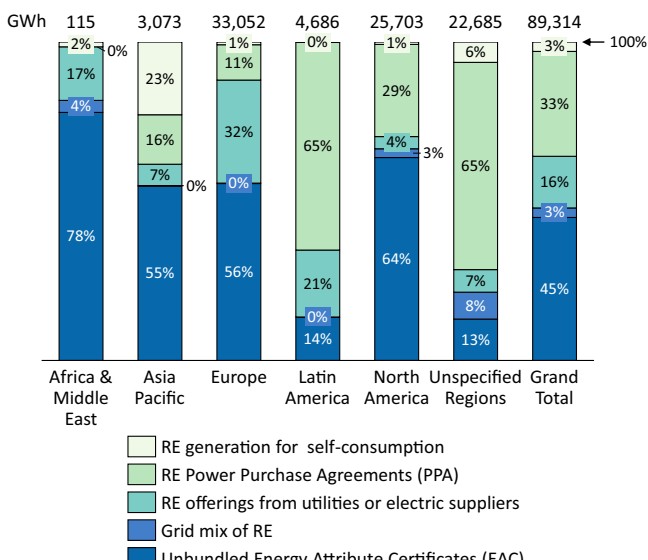

**Fig. 4 | Renewable electricity sourcing strategies by region.** Companies report sourcing strategies for 97% of the procured RE (89 of 92 TWh). We were able to allocate 67 TWh (or 75%) of the procured RE with a reported sourcing strategy to regions, leaving 23 TWh with unspecified regions. For example, if a company reported one sourcing strategy for several countries in different regions without specifying shares, the electricity was allocated to 'unspecified regions'.

three in more than one country, pointing out the unequal potential to impact the energy transition across companies.

Figure 4 shows the regional split between five sourcing options that RE100 companies report (see Supplementary Table 3): (i) RE generation for self-consumption, (ii) power purchase agreements (PPA), (iii) renewable electricity offerings from the utility, (iv) grid mix and (v) unbundled energy attribute certificates (EAC). Generally, the impact of procured RE on the energy transition is more likely additional to the transition under current policies and transformative if it is sourced locally[24,25]. The literature mainly refers to additionality as either additional to a no policy counterfactual, or additional to a current policy scenario, or additional to a pledged scenario (e.g., reaching the NDCs)[8]. In this analysis, we use the second concept, namely additional to current policies. By definition, generation for

self-consumption is additional to the transition of the broader electricity system as outlined in countries' policies. Procuring RE via local PPAs is also more likely additional compared the other strategies but the RE assets may be counted towards the achievement of national RE targets despite selling to one private off-taker exclusively. Such arrangements also create direct exposure of the procurer to the local policy environment, which makes it more likely that such a procurer intervenes to ease regulatory barriers to RE deployment and to implement ambitious RE policies. Therefore, they are more likely transformative. The other sourcing strategies may still be partly additional to current policies and transformative as they build interest groups in favour of a rapid transition towards RE, but we argue that there is a ranking of sourcing strategies in terms of additionality and transformational potential from (i) to (v).

While self-consumption, the most direct and local sourcing strategy, is used only for 3% of global demand, it makes up almost one-quarter in the Asia Pacific region (though overall RE procurements in the region are low). More importantly, PPAs are being used for roughly one-third of RE100 demand, with a very high prevalence in Latin America (65%). RE offerings from utilities, by contrast, are most prevalent in Europe (32%), possibly because European utilities have more such products on offer. Procuring RE directly via a green grid mix is seldom used in the reporting of RE100 companies, whereas the least direct sourcing option, unbundled EACs, is the most prevalent sourcing strategy (45% of global demand) in all regions but Latin America. Figure 1 in the Supplementary Information shows sectoral differences in sourcing options. We observed that self-consumption is concentrated in the manufacturing sector, which is evident given the often electricity-intensive and concentrated production facilities in the sector. Furthermore, we see that PPAs and RE offerings from utilities are used in all sectors except apparel and transportation. These two sectors mainly rely on EACs, which are also prevalent (68%) in the food, beverage and agriculture sector. In sum, 36% of RE100 demand likely has a direct, local, and additional impact because self-generation or sourcing via PPAs implies a physical link and local electricity generation. A caveat to this is the inclusion of virtual PPAs in RE100 data. In such PPAs, the off-taker (e.g., the RE100 signatory) assumes the market risk from an RE generator and receives unbundled certificates in exchange. A further 19% of RE100 demand likely has a direct and local effect because sourcing via RE offerings or grid mix implies a physical link and usually local generation (unless utilities are able to offer RE by purchasing certificates themselves). However, the additionality is less clear because RE100 demand likely incentivizes utilities to increase RE

generation capacities, but it may also replace existing RE demand. Lastly, 45% of RE100 demand has an indirect impact at most because EACs neither involve a physical link, nor a clear additionality. At least, because the electricity has to be sourced within market boundaries, RE100 ensures a more robust link compared to "pure" certificates. RE100 stipulates that unbundled RE procurement must happen within electricity market boundaries, defined as a common regulatory framework with sufficient interconnectors and a mutual recognition of certificates between suppliers. The specifications of electricity market boundaries have changed over time. Currently the US and Canada form one and most EU countries form another one[26].

### Dynamic and scenario-based impact

A Paris-compatible climate pathway requires the rapid acceleration of RE deployment, particularly over the next decade. Accordingly, the RE100 initiative encourages its members to set a first interim target no later than 2030. However, 113 of the 185 companies reported no interim targets before 2050. The omission of interim targets gives corporate members the opportunity to leave their electricity mix unchanged until the target year in which they ramp up RE procurement to reach their target. We model this behaviour, called 'stepwise' and contrast it with a 'linear' target achievement trajectory in which companies move linearly towards achieving their targets (final or interim). The former is what companies committed to in RE100, and the difference is stark. Figure 5 shows the resulting temporal dynamics from 2020 to 2030 to shed light on near-term dynamics because the typical corporate planning horizon spans less than 10 years. This shows that a linear trajectory would lead to higher RE procurement by RE100 in all years (lines) with higher growth rates (stacks) in all years but the common target years being 2025 and 2030.

In the linear trajectory, RE100 companies that communicated no interim target, but only a final (e.g., 2050) target, are assumed to increase their RE share linearly every year. In contrast, in the stepwise trajectory, companies only increase their RE share in target years. Note that both trajectories show an increase of RE demand in every year because companies' absolute electricity demand is expected to grow in line with growth rates in the countries, they have business operations in (assuming the Stated Policies Scenario by the IEA, see Methods). Hence, even if the RE share remains constant, the absolute RE demand increases slightly. The result of the lack of an interim target pathway is drastic. In just 10 years between 2020 and 2030, we estimate an RE demand lag of 361 TWh if RE100 members 'only' reach their

targets without linear progression towards them. Put into an emissions perspective, it would translate into savings of 14 Mt $CO_2$ per year if RE replaced European gas-fired power plants or 33 Mt $CO_2$ if it replaced coal-fired power plants (emission factors from ref. 15). This range roughly represents the annual $CO_2$ emissions from Lithuania (lower bound) or Singapore (upper bound).

Contrasting 2030 with 2020, Fig. 6 disaggregates the results for the 20 countries with the largest estimated RE demand by RE100 companies in 2030. We first observe that the absolute RE demand by RE100 companies grows in every country from 2020 to 2030 (see Fig. 6a). On the one hand, progress towards RE targets by RE100 companies contributes to this, while on the other hand, overall electricity demand growth also contributes. Second, Fig. 6b shows that the share of RE sourced by RE100 companies develops unevenly across countries from 2020 to 2030. If the share increases over time, it indicates that RE100 companies with local operations in a given country deploy RE faster than the country does as a whole (see Methods for country projections). If the share decreases over time, it indicates that the country is more ambitious in deploying RE than RE100 companies that are active locally. We find that seven, mainly Western and industrialised countries (the Netherlands, the United Kingdom, the United States, Germany, Switzerland, France and Sweden; ordered by size of share change), are more ambitious than locally active RE100 companies. We also find that RE100 companies are more ambitious than 13 countries. With some exceptions, such as Japan and Australia, these are mainly emerging economies (e.g., Argentina, Malaysia, Mexico, Thailand, and South Africa).

Beyond procuring additional RE, one of the crucial potential impacts of initiatives, such as RE100, lies in their potential impact on policy dynamics. If companies with operations in a given country commit to ambitious targets, such initiatives may encourage more ambitious public policy. Countries where RE100 demand for RE grows faster than domestic RE deployment (i.e., the mentioned emerging economies) may be candidate countries. Figure 7 provides an analysis of the potential impact of RE100 on national policymaking. It plots the national regulatory environment (RISE indicator by the World Bank, see Figure caption) against the share of domestically available RE that RE100 companies will need in 2030 (Fig. 6b).

We observe four things. First, RE demand from RE100 companies on average accounts for only 2% of domestically available RE in 2030. Yet, many companies will decide to source RE from abroad (see Fig. 4), which means that the impact of RE100 on domestic policymaking will

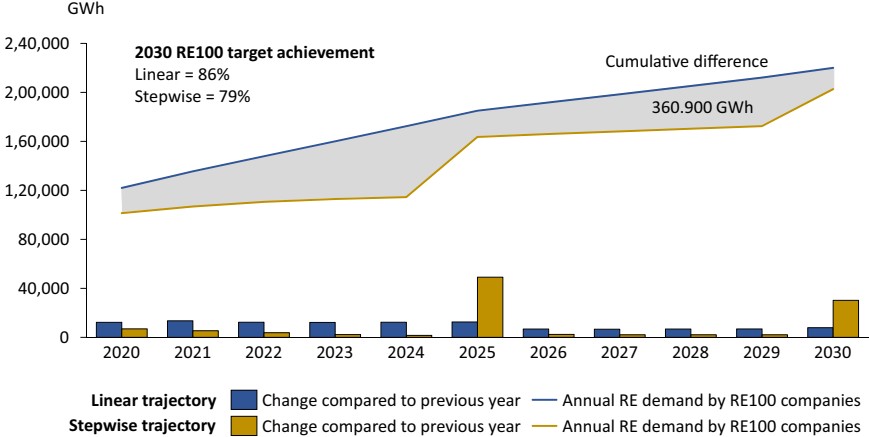

**Fig. 5 | RE demand caused by RE100 from 2020 to 2030, assuming that all RE100 member companies reach all interim and final targets.** The linear trajectory assumes that companies progress linearly towards achieving their RE targets. Stepwise trajectory assumes that companies procure the required RE to reach their RE target only in the target year. Shaded in grey is the cumulative demand difference between linear and stepwise trajectories. Note that the linear and stepwise demand would be equal once all RE100 companies reached 100% RE in 2050. See Methods for assumptions about country-level electricity demand evolution (identical between linear and stepwise trajectories).

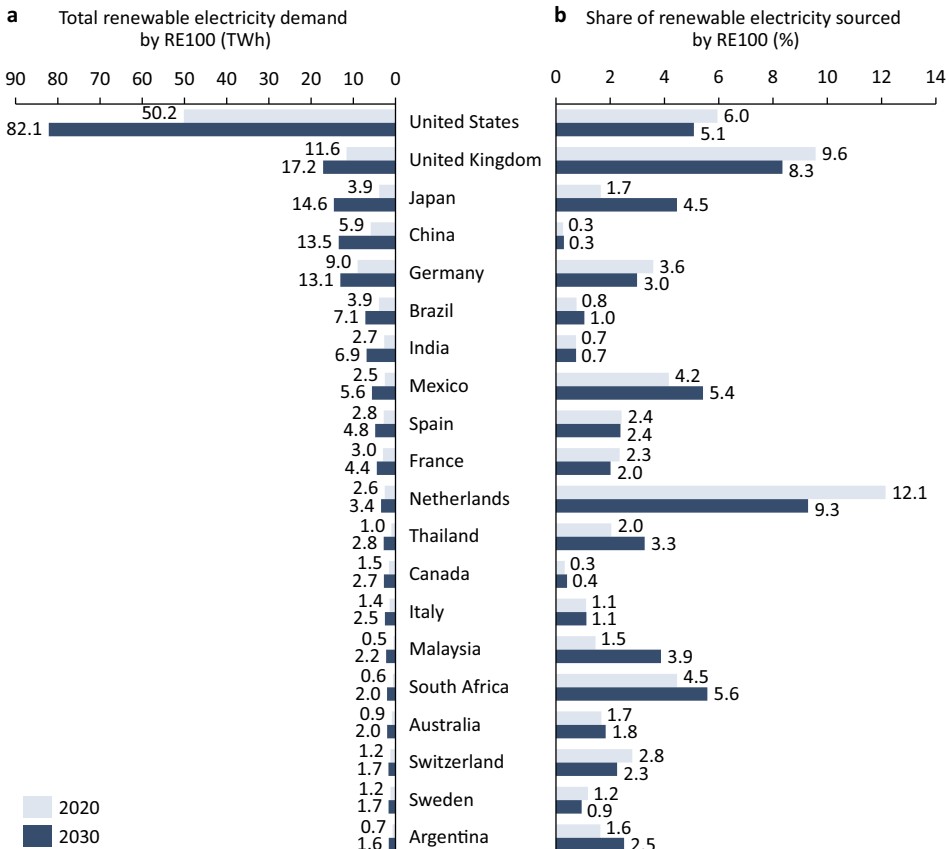

**Fig. 6 | Renewable electricity demand by RE100 companies in 2030 versus 2020. a** Estimated RE demand by RE100 companies in 2020 and 2030. **b** Estimated RE demand by RE100 companies in percent of estimated total domestic RE generation in 2020 and 2030. Projected domestic RE generation is based on the Stated Policies Scenario by the International Energy Agency (see Methods). The panels cover the 20 countries with the largest estimated RE demand by RE100 companies in 2030.

likely remain marginal in most countries. Second, we see that the policy impact potential of RE100, denoted by countries facing above-average RE100 impacts in 2030, is concentrated in high income countries (which also tend to have more ambitious policies, though this relationship is less clear). While high-income countries tend to have more ambitious RE targets compared to other income classes, the global concentration of the economic activity makes corporate initiatives more relevant in these countries. Third, we observe that 12 countries in quadrant A are most likely to face policy bottlenecks and associated lobbying efforts to improve the policy framework for RE by RE100 and its members. Some of these countries, e.g., Bahrein (BHR), Saudi Arabia (SAU), and Singapore (SGP), will see RE demand from RE100 exceeding 10% of the domestically available generation (15%, 25%, and 34%, respectively). Fourth, we find that 18 countries in quadrant B, predominantly European, will also face proportionally high RE demands by RE100 members in 2030. While these countries have better policies in place to facilitate RE deployment, they may face other constraints to furthering their RE generation, namely space or political opposition. For the latter, corporate initiatives may be important to tilt the interest group balance in the political process towards RE deployment.

These impacts remain a conservative estimate because they assume that RE100 stays constant in size. We therefore complement the analysis with two growth scenarios to illustrate the potential of corporate initiatives. Note however that the additionality of an expansion is unclear because companies may have implemented equally ambitious RE procurement policies without joining RE100. We discuss this challenge in the final section on future research avenues. The 'membership expansion' scenario represents ambitious but

realistic membership growth. We assume that RE100 membership grows in line with previous growth rates and reaching 1000 member companies by 2030 (see Methods for details). The scenario 'more ambitious targets' represents a tightening of the overall RE100 target. We assume that RE100 member companies will reach 100% RE by 2030 instead of 2050. Given the cost competitiveness of RE across a wide array of geographies[27] and the urgency to decarbonise this decade[1], such a scenario may be necessary if RE100 companies wish to position themselves as ambitious actors in the energy transition. We deliberately abstain from modelling a combined scenario, because we aim to contrast the relative importance of the two dimensions.

Comparing the scenarios in Fig. 8, we gain three main insights. First, in the baseline scenario, RE100 companies are expected to source precisely the same share of globally available RE in 2030 as they did in 2020 (1.6%, up from 1.4% reported in 2018, see Supplementary Table 1). Hence, RE100 companies are not more ambitious in expanding their RE procurement than countries on the global average. Second, increasing the ambition to 100% RE in 2030 only marginally changes the picture. We found that in the 'more ambitious targets' scenario, RE100 companies would only procure an additional 36 TWh of RE by 2030 (254 TWh compared to 218 TWh in the baseline scenario). This would translate into a 1.9% share of globally generated RE instead of a 1.6%. Hence, increasing the ambition on the timeline to reach 100% RE does not substantially expand the role of RE100 in the global energy transition until 2030. Third, we found that expanding RE100 to cover additional companies has a substantial impact. By 2030, expanding RE100 to 1000 members would lead to an approximately five-fold increase in RE procurement by RE100 companies (1,035 TWh compared to 218 TWh in the baseline scenario). In such a

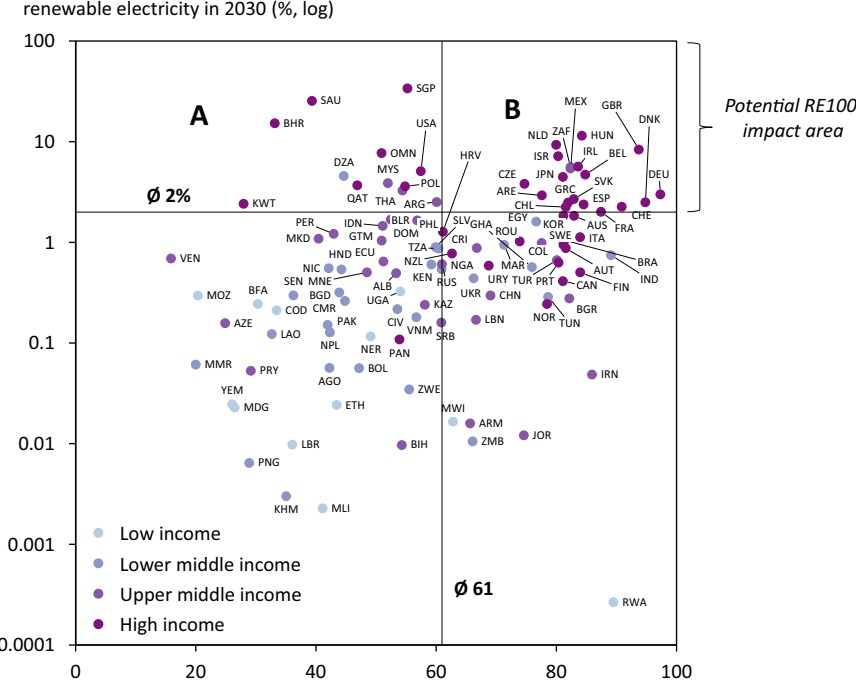

**Fig. 7 | Potential RE100 policy impact areas by 2030.** The *x*-axis shows the RISE−Regulatory Indicators for Sustainable Energy from the World Bank in 2019 (latest available year), the *y*-axis shows the estimated RE demand by RE100 companies in 2030 as a share of estimated domestic RE supply in 2030 (see Fig. 6b). The vertical and horizontal lines denote the respective unweighted averages across countries. A and B denote quadrants as discussed in the text. Income classification are as per the World Bank. Figure shows the 111 countries, for which data on both axes is available.

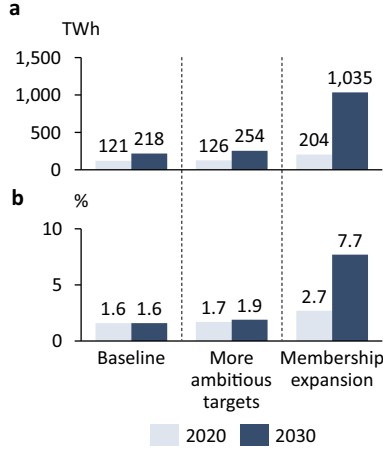

**Fig. 8 | Global RE100 effects for the three scenarios. a** Estimated global RE demand by RE100 companies in 2020 and 2030 (in TWh). **b** Estimated global RE demand by RE100 companies in percent of the estimated total RE generation in 2020 and 2030. See Methods for details on the scenarios.

'membership expansion' scenario, RE100 companies would procure 7.7% of globally projected RE by 2030, thus potentially having an accelerating impact on the energy transition. All scenarios feature a switch from conventional electricity generation technologies to RE between 2020 and 2030. Using the conservative assumption that the additional RE replaces a fuel with the average emission intensity of European gas-fired power plants[15], RE100 will contribute an emission reduction of 41–349 Mt $CO_2$ per year (baseline and membership expansion). Despite methodological differences these aggregate

numbers align with existing overall estimates[28]. Over a period of ten years, this translates into 0.1–1.3% of the $CO_2$ budget available at the end of 2022 to stay within 1.5 °C of global warming.

## Discussion

While previous studies and reports of corporate renewable energy initiatives have used aggregate numbers to discuss their impact[15,16], this study analyses the potential impact of a corporate RE initiative on the country level. In doing so, we make three key contributions. First, this is the first scientifically rigorous and transparent assessment of RE100, allowing for an evidence-based assessment of the climate impact this corporate initiative may or may not have. Second, we estimated the impact of individual RE100 companies on the country level, which can be replicated in the future and will likely become more precise as granular data becomes more available. Third, the country level results make it possible to identify countries where RE100 potentially has a transformative impact beyond sourcing additional RE by influencing policy strategically. Generally, our study dampens the hope that corporate RE initiatives will have a substantial impact on accelerating the energy transition. Globally and on average, RE100 companies' decarbonisation plans are just as ambitious as the countries they are based in, such that the share of RE sourced by RE100 companies hardly increases 2020–2030 in a baseline scenario. It seems that the potential (modest) impact of RE100 on the energy transition lies in some emerging economies, where the procurement plans of RE100 members are more ambitious than the RE generation and targets of the country as a whole. However, the quantification of whether demand from RE100 companies is additional and leads to a build out of RE capacities that would otherwise not have happened remains a limitation of this study.

In this paper, we suggest three ways RE100 (and other similar initiatives) can increase their impact. First, these initiatives need to enforce meaningful and ambitious interim target pathways to avoid

late investment by laggard member companies just before the (long-term/interim) target year. As shown in this study, enforcing a roughly linear progression towards existing targets would lead to an additional 361 TWh of RE procured by RE100 companies between 2020 and 2030.

Second, RE100 should increase their ambitions in terms of sourcing strategies. Our analysis shows that EACs are the most common RE sourcing strategy used by RE100 members. While this may be the cheapest available option for companies, these EACs can be double counted (as corporate and public effort) unless well-regulated. Switching to other sourcing strategies would allow companies to assure that their RE procurement has a more direct impact on accelerating the energy transition without depending on stringent EAC regulations. To this end, RE100 should not only collect data from but also offer best practice sharing among its members. For instance, some companies in the IT services sector (e.g., Alphabet/Google) are building knowledge on how to source RE that match the demand locally and 24/7[24], while other sectors are only at the beginning of global RE procurement. Fostering knowledge spillovers could be a valuable contribution of corporate initiatives.

Third, initiatives such as RE100 can take a convening role and orchestrate corporate interests towards more conducive RE policies in countries with less ambitious decarbonisation policies for the electricity sector. By gathering detailed country-level insights into RE demand patterns of member companies, RE100 could strategically identify bottlenecks and engage with governments in these countries. Such policy impacts are particularly interesting in countries, where current RE policies are not stringent and where vested interests around fossil fuels are prevalent, e.g., Bahrein, Kuwait or Saudi Arabia (see Fig. 7). Furthermore, bottlenecks may also depend on the sourcing strategy. For example, moving from EACs to PPAs requires an adequate regulatory framework to allow for PPAs and to provide the necessary regulatory certainty to engage in such long-term contracts. Using this influence to shape more ambitious RE policies may be one of the largest levers available to corporate energy and climate initiatives, such as RE100.

Finally, we believe that further research is needed on corporate climate and clean energy initiatives, particularly more effort is required to quantify the (potential) effect they have in addition to government commitments. Two areas seem of particular importance, namely analysing location-specific data on sourcing strategies (see Fig. 4) and evaluating counterfactuals (e.g., how would companies have procured electricity without joining RE100?). However, compared to governments, companies are more restrictive in sharing data for competitive and strategic reasons; it is likely that quantitative analyses will always require a degree of estimation and assumptions. This study presents the first attempt at the case of renewable electricity procurement on which subsequent research can build. Only rigorous potential impact analyses allow the public to judge which corporate initiatives are potentially contributing to the clean energy transition and climate change mitigation, thus helping fill the vast gap in the implementation of governmental climate pledges.

## Methods
### Scope of analysis
The RE100 initiative requires member companies to achieve 100% renewable electricity (RE) by 2050 at the latest, with interim targets of 60% by 2030 and 90% by 2040, to report data on their progress annually through the RE100 reporting template or CDP's disclosure platform and to engage other businesses to join them (reference to CDP Worldwide is abbreviated to CDP in this paper)[29]. As of December 2019, RE100 had 211 member companies. We searched for data on all these companies following the process outlined below, and we were able to find electricity demand data for 185 companies. Our total estimated electricity demand for these 185 companies is 227 TWh, which corresponds precisely to what RE100 reported in December

2019[30]. The reference year for all data is 2018, unless indicated otherwise. If companies reported in fiscal years instead of calendar years, we chose the most recent available fiscal year, which covers 2018. We allocated companies to sectors according to RE100, which uses the CDP Activity Classification System (see Supplementary Table 4 for an overview). We used region and income classifications from the World Bank for regional and income splits and summarised regions by combining 'Middle East & North Africa' and 'Sub-Saharan Africa' to 'Africa & Middle East' and combining 'East Asia & Pacific' and 'South Asia' to 'Asia Pacific'. 'Europe' includes Central Asia and 'Latin America' includes the Caribbean.

Where we have reported data on a country level, we have excluded countries and territories with population below 1 million in 2018[31], as most of them are island states or finance-dominated territories for which our allocation proxies (see below) are less suited. Therefore, we excluded the following 23 countries/jurisdictions: The Bahamas, Bermuda, Barbados, Brunei Darussalam, Curacao, Cayman Islands, Dominica, Fiji, Micronesia Fed. Sts., Gibraltar, Isle of Man, Iceland, Liechtenstein, Luxembourg, Macao SAR China, Monaco, Marshall Islands, Malta, Montenegro, Sint Maarten (Dutch part), Seychelles, British Virgin Islands, Virgin Islands (US). Accordingly, country-level figures and maps show the (renewable) electricity demand of RE100 companies in 129 of the total 152 countries in which these companies report activities. These 129 countries covered 99.9% of the TE demand by RE100 companies in 2018.

### Allocating RE100 electricity demand to countries
In this step, we use a variety of data sources to allocate RE100 electricity demand to countries. Data processing and analysis were conducted in Microsoft Excel. Figure 9 shows a flowchart of the process that describes the data collection. Throughout the process, we followed the principle of seeking the most specific data available (e.g., electricity dominates energy data) at the most granular level (e.g., country dominates regional data).

We first collected TE demand for each company using the CDP climate change questionnaire (154 of the 211 companies reported via CDP) and further publicly available information like annual reports, social responsibility reports and websites. The CDP climate change questionnaire is filled in by participating companies via an online reporting system. It collects information on climate risks, exposure to these risks and opportunities alongside general corporate information (see Data availability for more details). We mainly use section 7 (emission breakdown) and section 8 (energy). In cases where the retrieved data is insufficient or unclear, we reached out to the companies individually via email. For 12 companies, we were unable to find 2018 data and used the closest available year between 2016 and 2019. Then, we used the data from ORBIS (Bureau van Dijk Electronic Publishing Ltd.) to determine the countries in which a company reports activity. Specifically, we defined a company as being active in a country if there was a subsidiary reporting either employees or revenue figures and if, for the available figures, the subsidiary employed at least 10 employees and reported revenues of at least USD 1 million.

Based on this, we identified 30 companies with activities in only one country (domestic market only). For these companies, we simply used the TE demand and allocated it to this country ($N = 30$, see dataset 1 in Fig. 9). If a company was active in more than one country, we proceeded to collect further data to allocate electricity demand to other countries.

For the remaining 155 companies with a multinational presence, we screened two different data sources to allocate electricity demand to countries. First, we searched for energy (instead of electricity) data at the country level reported via CDP or company reports because companies do not disclose country-level electricity demand. If data was available, we assumed that the country-level energy demand shares equalled the electricity demand shares and allocated electricity

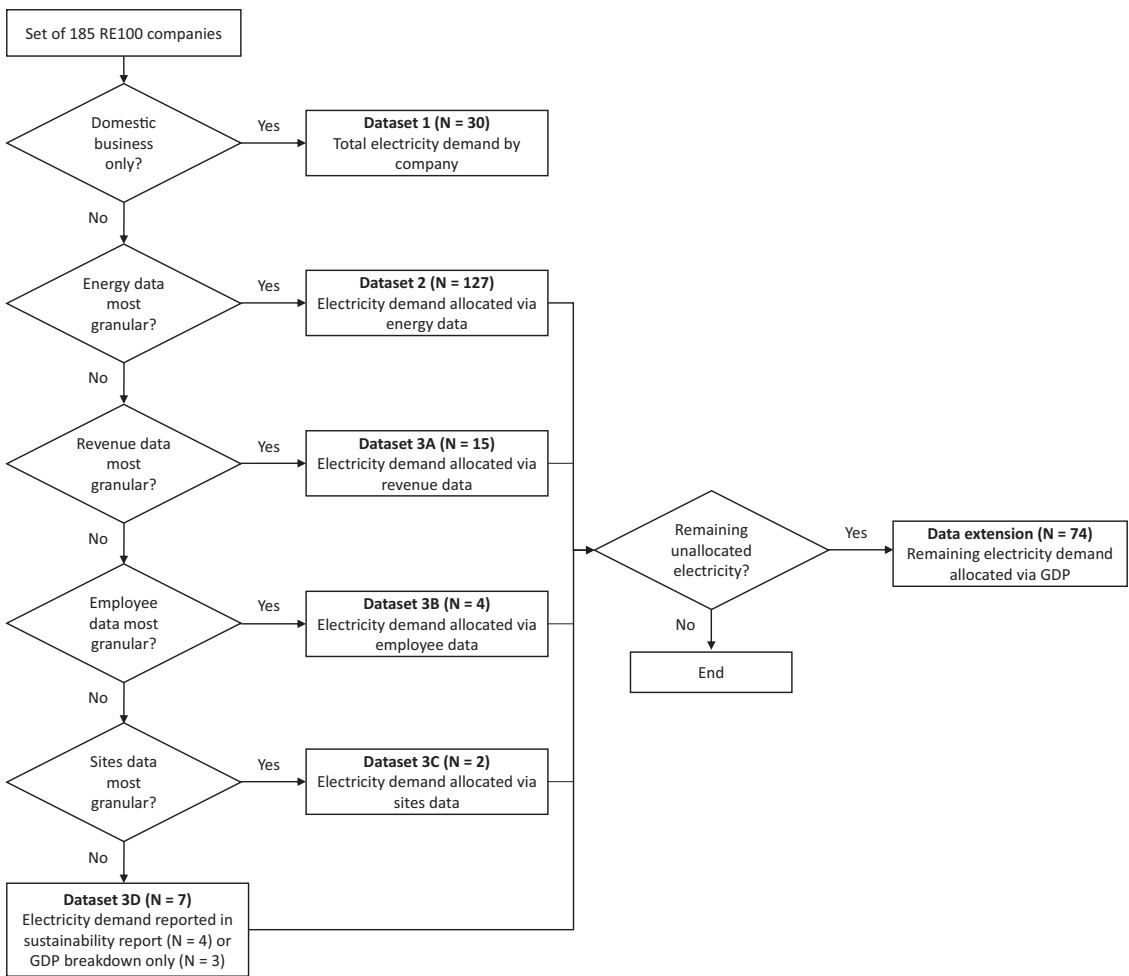

**Fig. 9 | Allocation of RE100 electricity demand to countries.** The diagram shows the decision tree to allocate the electricity demand of each company to countries.

demand accordingly (see dataset 2 in Fig. 9). We used energy data from the CDP climate change questionnaire (item C8.2a) by adding up the consumption of the four components, namely electricity, heat, steam and cooling, as defined by the Greenhouse Gas Protocol Scope 2 Guidance. Note that reporting energy data at the regional or country level is a requirement as part of the Scope 2 emissions breakdown in the 2019 CDP questionnaire – hence, some companies do not report country-level energy data. For 67 companies, we were able to allocate all electricity demand to countries using energy data. For 42 companies, we found country-level energy data for some but not all countries of activity, and for 18 companies, we found regional energy data only (still the most granular data source). In these cases, the dataset extension was used (see Fig. 9). Second, we searched for revenue, employee or site data in ORBIS, company reports and web resources. In case one of these proxies was used to allocate electricity demand to countries, we followed the same logic of transferring proxy shares to electricity demand shares as described previously (see datasets 3A, 3B and 3C in Fig. 9). For each company, we chose the variable offering the highest granularity. If all four variables offered the same level of granularity, we preferred energy over revenue over employee over site data. For example, H&M reported energy data for Sweden (3.8%) and the rest of the world (96.2%) to CDP. In its annual report, the company reported revenue data for 58 operating countries, of which Sweden accounted for 4.1%. In this case, revenue data offered the highest granularity, and we allocated the electricity demand reported by H&M to countries proportionally based on the reported revenue figures.

For 74 companies, we were unable to allocate all reported electricity data to countries using the four variables described previously.

In these cases, we used GDP data to allocate the electricity demand to the remaining regions (see data extension, Fig. 9). Four companies reported detailed electricity data in their sustainability reports for three companies, McKinsey, Elopak and NREP, we were unable to find any data and allocate all electricity demand to the countries of operation using GDP data (see dataset 3D, Fig. 9). However, not all electricity demand could be allocated to countries. For example, if a company reports electricity demand in Africa via the CDP questionnaire, but ORBIS indicates no subsidiaries in Africa, the reported electricity remains unallocated. The same applies to electricity demand, which is reported for Jersey and Guernsey, which are not listed in the World Bank country list. We used GDP data in current USD for 2018 as of July 2020 from the World Bank[32] and supplemented it with UN data[33] to increase country coverage (note that the GDP data for Taiwan is from the Taiwan Statistical Bureau).

We applied consistency checks and a four-eye principle to ensure the accuracy of the data. After the collection and allocation of electricity demand to countries by one researcher, the other researcher double-checked the results by going through the entire allocation procedure again. Where issues in the allocation were identified, they were discussed among the researchers. Note that for the total energy demand by company from the CDP questionnaire, we had the option to take aggregate reported figures (top-down) or we could aggregate the figures obtained from regional or country-level reporting (bottom-up). Although these should be equal theoretically; there is often a slight difference in practice. If the bottom-up aggregation was larger, there was no issue. If the top-down sum was larger, the difference between top-down and bottom-up remained unallocated to countries.

**Table 1 | Scenario description.** Note that in the membership expansion scenario, new member companies match the average existing RE100 company (e.g., size, geographic activity, targets)

|  | Electricity demand growth | RE deployment | RE100 |
|---|---|---|---|
| Baseline | Stated policies scenario STEPS, linear interpolation[36] | STEPS, linear interpolation | Interim and final targets met, linear interpolation |
| More ambitious targets | STEPS, linear interpolation | STEPS, linear interpolation | Interim and final targets met, linear interpolation<br>All RE100 members achieve 100% in 2030 at the latest |
| Membership expansion | STEPS, linear interpolation | STEPS, linear interpolation | Interim and final targets met, linear interpolation<br>Membership grows linearly to 1000 in 2030, linear growth |

This difference was included in the total, but it does not appear in the country-level breakdown. It amounts to 0.6% of the total estimated RE100 company electricity demand and therefore remains negligible.

### Estimating TE and RE generation by country

After allocating RE100 electricity demand to countries for each member company, we had to estimate TE and RE generation by country. Combining the two yielded the required country-level data for RE100 electricity demand in relation to domestic generation.

We used TE generation in 2018 from the United Nations Energy Statistics Database as per July 2020 (net electricity generation by country)[34]. RE generation is based on the International Renewable Energy Agency (IRENA) Renewable Energy Statistics[35]. To project TE generation and RE generation into the future, we relied on data from the International Energy Agency (IEA)[36]. We took projections based on the continuation of current policies (Stated Policies Scenario) and interpolated linearly between 2018 and the target years 2020, 2025 and 2030 using the most granular available projection in the IEA dataset. For Brazil, China, India, Japan, the United States and Russia, country-level projections were available, and for the remainder of countries, we used regional projections (Africa, Asia Pacific, Central and South America, Eurasia, Europe, Middle East, North America and Southeast Asia).

Hence, for countries where country-level projections were unavailable, the TE demand in 2030 was given by Eq. (1).

$$TE_{c,2030} = TE_{c,2018} + \frac{TE_{r,\Delta}}{TE_{r,2018}} \times TE_{c,2018} \qquad (1)$$

where subscript $c$ denotes country-level variables, subscript r denotes regional variables and subscript $\Delta$ denotes the difference between 2030 and 2018. The RE demand by country was then given by Eq. (2).

$$RE_{c,2030} = RE_{c,2018} + \frac{RE_{r,\Delta}}{TE_{r,\Delta}} \times TE_{c,\Delta} \qquad (2)$$

where we imposed the boundary condition that $RE_{c,2030} \leq TE_{c,2030}$ to arrive at the projected 2030 RE demand for each country. We then assumed that a company's electricity demand in a given country develops in line with its TE demand.

### Defining RE100 progression path towards targets

In the next step, we defined how RE100 companies achieve their targets on a time scale. We assume that companies' total electricity demand grows in line with electricity demand growth in the countries they operate in (see two previous sections). Our baseline assumption was that all RE100 companies achieve their interim and final RE targets, and they progress towards these targets on a linear pathway. Hence, we interpolated RE shares linearly between the targets for each company. However, because RE100 companies may be able to adapt their electricity sourcing reasonably quickly, we also used a stepwise target achievement, where we assumed that companies remain inactive until

the year before a target (interim or final) and adapt their electricity sourcing to meet the target in the target year. Except Fig. 5 in the main text, which exemplifies the difference between the two progression paths, we have reported the linear progression path. Irrespective of the path, we assumed that each company progresses towards its goal simultaneously in each country of activity; hence, the achieved RE shares will always be identical across all countries of activity for a given company.

### Defining RE100 scenarios

Finally, we defined three scenarios for the development of RE100, as shown in Table 1. Overall electricity demand growth and RE growth were calculated as described in Step 2 for all scenarios. Baseline assumes that RE100 membership stays constant and that all member companies achieve their targets, as described in Step 3. 'More ambitious targets' scenario assumes that RE100 companies raise their ambition to achieve 100% RE by 2030 at the latest. Companies that already had an earlier 100% target retained those targets. The purpose of this scenario is to illustrate the impact of corporate RE procurement if RE100 responded to the widespread criticism that 2050 targets are inadequate, given the most recent IPCC assessments of necessary GHG emission reductions to stay in line with the Paris Agreement targets[1]. 'Membership expansion' scenario reflects that RE100 grows in line with its communicated targets, analogous to scenarios proposed in the literature[15]. RE100 targeted membership was 1000 in 2020 and 3000 in 2030[37] but by 2021 the initiative counted only 300 members. Hence, we assumed continued membership growth to 1000 by 2030, which reflects realistic growth targets. Note that membership growth may be much larger due to rapidly falling RE costs, but additional member companies will likely be smaller in size compared to existing member companies. We assumed that additional member companies reflect the average size, the average geographical activity distribution and the average RE100 target (level of ambition) of current RE100 members.

### Reporting summary

Further information on research design is available in the Nature Portfolio Reporting Summary linked to this article.

## Data availability

Data to reproduce the figures is available on Figshare (https://doi.org/10.6084/m9.figshare.23496956). The underlying company–country electricity demand matrix (allocation of electricity demand from company to countries) and other more granular data is partly based on data from CDP (combined with data from other sources), and reproduction of the CDP data by any third party is forbidden by CDP license terms. The data from CDP will be made available by the corresponding authors upon reasonable request given the requesting party has the permission from CDP. For the full list of available questions and version control of the 2019 CDP climate change questionnaire, see: https://guidance.cdp.net/en/guidance?cid=8&ctype=theme&idtype=ThemeID&incchild=1µsite=0&otype=Questionnaire&tags=TAG-646%2CTAG-605%2CTAG-600.

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

## Acknowledgements

David Gut provided research assistance. We thank the members of the Energy and Technology Policy Group and the Climate Finance Policy Group for their feedback on earlier drafts of the manuscript. The authors declare no specific funding.

## Author contributions

F.E., B.S. and T.S. developed the research idea and design; R.Z. and V.H. collected and prepared the data; F.E. and R.Z. analysed the data; and F.E. wrote the paper with inputs from T.S., B.S. and R.Z.

## Competing interests

The authors declare no competing interests.
