## [Peer Review File · Nature Communications]

REVIEWER COMMENTS

Reviewer #1 (Remarks to the Author):

The analysis of the RE100 initiative, downscaling the targets to country level is well performed, and the article is also clearly written. For me, the question is whether this "represents important advances of significance to specialists within the field". The main conclusion is that companies should focus on interim targets and RE on expansion. This is not new. Would you be able to increase the policy relevancy of your results? Or frame it that although this is being mentioned in the policy arena, no scientific assessment are done thus far. See for example

<https://www.net-zero-hub.com/net-zero-target/interim-emissions-reduction-targets/>

<https://unfccc.int/climate-action/race-to-zero-campaign#eq-3>

One general remark that I miss in the analysis: could you elaborate what the impact of RE100 expansion? Under which conditions does this result in higher uptake of renewable electricity? You could also say that is just replaces RE increase somewhere else

More specific comments

- P2, l29, 'most of these pledges concern long-term future'. Most commitments are in the NDCs, which are mainly for 2030
- P2. l39, you refer to Hsu et al. (2019) and Kriegler et al. (2018). I do not think these article debate the actual contribution to climate change mitigation. At most, they raise the question if these climate actions would result in additional reductions to national pledges or policies
- P2 l51, this also holds for renewable electricity from off-shore wind or large-scale PV fields. Main question is to what extend the RE100 initiative does result in additional reductions. If companies procure renewable electricity, does this replace renewable electricity used by household, or add to it? If for example Microsoft procures 100% REN electricity for a data centre, does this replace household renewable electricity

- P3, l61/62 you seem to suggest you assess multiple initiatives, but this is not the case
- P3, l64. Is it possible to use one historical base year? You use 2018 here, but 2021 on p1/l47. Based on the IEA data, in 2019 total electricity production is 27,044 TWh (see link). 340 TWh of RE100 electricity (in 2021) would be 1.3%, not 0.01%

<https://www.iea.org/data-and-statistics/data-browser?country=WORLD&fuel=Energy supply&indicator=ElecGenByFuel>

- P3, l68. How can it already be higher in 2020 (which is the past) than current membership mentioned in line 64?
- P4, l80, I would say large instead of huge
- P5, l95, a more detailed comparison based on the supplementary information from (13) would be interesting
- P7, l143, what do you mean with 'direct'? Please clarify (in the text)
- P9, l180. The difference between linear and stepwise is an interesting analysis. It would even appeal more to the imagination if you could link it to a understandable number. How much is GWh for example in CO2 terms and/or percentage of 1.5C or 2C budget (400-1000 GtCO2). Or something else
- P10, l204-208. What data/projections did you use to determine national 2030 RE shares? This is important to add to the text. Also add this to the caption of Figure 6
- P11, l226-227, this is worded too strongly (.e.g. 'realistic'). See for example Figure 6.30 in the AR6 WGIII report. Even for a 1.5C scenario, the range of %-low carbon energy is between 70-85%
- METHODS

- P17, l343-350 Why did you use all energy data, if electricity only is also available in the CDP dataset, and this is what you are analysing in your research?

- P19, l402, this formula seems wrong. $TE_c,2030$ is both on the left side and right side, so can be crossed out against each other. Then the second term on the right is equal to zero. First term on the right side should probably be $TE_c,2018$

Reviewer #2 (Remarks to the Author):

This manuscript aims to answer an important policy question of whether corporate climate action initiatives such as RE100 would really deliver substantive mitigation contributions. This research work is timely, as there has been an increasing scepticism against corporate climate pledges following the COP26.

I find the quantitative analysis solid overall, but some additional work would be required before the manuscript could be accepted for publication. Below are a few high-level comments, followed by more detailed comments.

High-level comments

1. Readability: Due to the objective of RE100, the text goes back and forth between total electricity generation/demand and (total) renewable electricity generation/demand. This makes it difficult to read through the manuscript even for a person like me, who's more familiar with the topic than most other readers. I do not have a good suggestion, but perhaps good to have the revised manuscript checked by a scientific copyeditor (in case you haven't done it yet).

2. Key messages: In the Discussion section, you have the conclusion: "Generally, our study dampens the hope that corporate RE initiatives will have a substantial impact on accelerating the energy transition." To me this is the headline of your work, but the abstract is rather modest, only saying "raising doubts...". I also find a couple of sentences in the abstract to be potentially misleading or not worded right:

2.1. "We found that these companies source electricity in 129 jurisdictions and account for <0.01% of global electricity production" >> First, if it's about 129 jurisdictions, then wouldn't it be better to use the total electricity generation in the 129 jurisdictions as the denominator? Second, it seems that the <0.01% value is a close value of 0.009% in Line 93, but I'm not sure if the figure is correct (see under "Detailed comments" below).

2.2. “By 2030, stringent and frequent interim targets could lead to an additional 391 TWh of RE procured by RE100 companies” >> I understand this to be cumulative additional production between 2020 and 2030. If so, then it should be clarified as readers may mistake it for annual production in 2030.

3. Analysis: A policy report published last year also attempt to quantify the impact of RE100. Perhaps good to compare the overall results? <https://newclimate.org/2021/06/23/global-climate-action-from-cities-regions-and-businesses-2021/>

4. Analysis: I would like to see more in-depth analysis on the RE sourcing strategies presented in Figure 4. Some strategies contribute more to additional RE generation, while others are simply displacing fossil fuel-fired power generation to other consumers. Deeper qualitative insights into the RE sourcing strategies further strengthen your overall assessment that RE initiatives will have limited impact on accelerating the energy transition.

Detailed comments

1. L35: food companies can be carbon-intensive, depending on the emission scope you look at. If you consider the entire value chain including upstream land use-related emissions, then some food companies are one of the worst (see e.g. “Emissions Impossible” report by IATP).

2. LLL71: It should be clarified that the 2021 figure is the annual total.

3. L93: I understood the figure to be global. If so, then the figure might be an order of magnitude too small? Global electricity generation is about 27,000 TWh in 2019-20, so 221TWh would be a bit less than 1%. Or have I missed anything?

4. Figure 5: A cumulative difference of 360,900 GWh is presented, but does this correspond to the 391 TWh figure presented in the abstract and in L70? If yes, please explain the difference. (The Discussion section gives 361 TWh)

5. Figure 7: The Y-axis should be TWh, not TwH.

6. L299: “with interim targets of 60% by 2030 and 90% by 2040” >> I could not find these numbers in the cited report (RE100 annual report 2019: <https://www.there100.org/sites/re100/files/2020-09/RE100ProgressandInsightsAnnualReport2019.pdf>, and their two annexes). Please provide the page no. where they’re stated.

7. L394: Stated Policies Scenario

8. L395: WEO2021 has the historical data for 2019, so why 2018?

9. L402: The first term in the right side of the equation should be $TEc,2018$?

10. LL436-438: “We assumed that additional member companies reflect the average size, the average geographical activity distribution and the average RE100 target (level of ambition) of current RE100 members” >> (Clarification) Does this mean that the stepwise trajectory in Figure 5 will be scaled up by about factor 5?

11. Table M.1: STAPS >> STEPS

Reviewer 1

Comment	Response
The analysis of the RE100 initiative, downscaling the targets to country level is well performed, and the article is also clearly written. For me, the question is whether this "represents important advances of significance to specialists within the field". The main conclusion is that companies should focus on interim targets and RE on expansion. This is not new. Would you be able to increase the policy relevancy of your results? Or frame it that although this is being mentioned in the policy arena, no scientific assessment are done thus far. See for example https://www.net-zero-hub.com/net-zero-target/interim-emissions-reduction-targets/ https://unfccc.int/climate-action/race-to-zero-campaign#eq-3	We thank the reviewer for the positive overall comment and the constructive comments, which have helped us to improve the manuscript substantially. All changes are highlighted in yellow in the revised manuscript and we refer to line numbers in brackets. We agree that there is value in underlining the contribution of the paper more clearly. We believe that our analysis provides at least three key contributions: First, it is a scientific assessment of RE100 that goes beyond existing policy reports regarding rigour and transparency. The added value of this becomes clear when trying to compare our estimates with estimates in policy reports that tend to be methodologically unclear (cf. response to comment #3 of R2). Moreover, while policy reports have argued for interim targets, we propose a way of quantifying the effect of such targets and putting it into an emissions context (cf. response to a comment on the 1.5C carbon budget below). Second, this is the first attempt to estimate the impact of individual RE100 companies at the country level. To do so, we needed to develop a method based on publicly available data. Most existing papers either estimate global numbers (e.g., Lui et al., 2020 or RE100 reports) or limit the analysis to a select number of countries (e.g., Kuramochi et al., 2021). Third, it is this country resolution that makes it possible to a) develop a deeper understanding of the potential impact of RE100 on the energy transition and to b) identify countries where RE100 is more likely to have a transformative effect because the amount of RE likely sourced from RE100 companies in the near future exceeds available RE capacities. We have added an analysis to identify such countries systematically (new Figure 7) and discuss the implications in the revised manuscript. We have added these three contributions at the beginning of the discussion section in the revised manuscript (lines 319-325).

One general remark that I miss in the analysis: could you elaborate what the impact of RE100 expansion? Under which conditions does this result in higher uptake of renewable electricity? You could also say that is just replaces RE increase somewhere else	We thank the reviewer for this comment that raises relevant points. Indeed, the additionality of RE100 demand is difficult to examine because it requires counterfactuals. On the one hand, firms that are assumed to join RE100 in the expansion scenario could have stipulated equally ambitious RE deployment policies without joining the initiative. On the other hand, RE100 demand does not necessarily lead to new built capacities. Depending on the sourcing strategies applied, capacities can be added or existing RE capacities can be used to meet RE100 demand. We addressed this comment with three changes. First, we elaborated on the counterfactual when describing the scenario (297-281) and we explicitly refer to potential emission savings when RE replace gas-fired power plants (309-315). Second, we mentioned additionality as an avenue for future research (361-363), depending on data availability (e.g., sourcing strategies over time and location-specific). Third, we expanded the section on sourcing strategies (Figure 4) to be clearer about local impacts depending on the strategies, also referencing the newest RE100 guidelines on market boundaries for procurement (167-182).
P2, I29, 'most of these pledges concern long-term future'. Most commitments are in the NDCs, which are mainly for 2030	We thank the reviewer for this remark and we have added that second-generation NDCs are also inconsistent with a 1.5-degree pathway, referencing Iyer et al., 2022 in Nature Climate Change (28-30).
P2. I39, you refer to Hsu et al. (2019) and Kriegler et al. (2018). I do not think these article debate the actual contribution to climate change mitigation. At most, they raise the question if these climate actions would result in additional reductions to national pledges or policies	We thank the reviewer for this comment. Indeed, these references rather point to a) the need to assess the impact of non-state actors, such as business coalitions and b) the need for action beyond national policies to reach climate targets. We have replaced the references with Hale et al., 2020 in Climate Policy to make the point that evidence on non-state actor impacts is both scarce and urgently needed.
P2 I51, this also holds for renewable electricity from off-shore wind or large-scale PV fields. Main question is to what extend the RE100 initiative does result in additional reductions. If companies procure renewable electricity, does this replace renewable electricity used by household, or add to it? If for example	We thank the reviewer for this remark. We have added a sentence to be more precise (56-59). The main argument here is that corporate initiatives may help break the carbon lock-ins in countries, which rely on coal and natural gas for their electricity generation. Such lock-ins make large state-owned RE deployment or ambitious

Microsoft procures 100% REN electricity for a data centre, does this replace household renewable electricity	national policy difficult. The second point of this remark concerns additionality, which we address in several instances of the revised manuscript (please see response to the first comment). However, essentially, this remains a limitation of such a study as we will not be able to empirically assess or prove additionality. We now mention this explicitly in the discussion (331-333)
P3, l61/62 you seem to suggest you assess multiple initiatives, but this is not the case	We thank the reviewer for this comment. We have revised the manuscript accordingly.
P3, l64. Is it possible to use one historical base year? You use 2018 here, but 2021 on p1/l47. Based on the IEA data, in 2019 total electricity production is 27,044 TWh (see link). 340 TWh of RE100 electricity (in 2021) would be 1.3%, not 0.01% https://www.iea.org/data-and-statistics/data-browser?country=WORLD&fuel=Energy supply&indicator=ElecGenByFuel	We thank the reviewer for this remark regarding the base year and percentages. First on the year. We are constrained to using data from 2018 due to the availability of corporate reports and CDP filings at the time of data collection. In the meantime, RE100 has released more recent aggregate data, which is why we decided to mention the most up to date numbers (2021) in the introduction. We have taken care to be more precise on this point (e.g., 67). Second on the percentages. Indeed, this is a typo that should not have happened. It should be <1% of total electricity production, not <0.01%. We have corrected the mistake, and diligently re-checked the entire manuscript to make sure all numbers and percentages are correct. To explain the numbers in detail: We take net electricity production data by country from UN Statistics (see ref. 31). For 2019, UN Statistics reports a global production of 25,214 TWh, hence roughly comparable to the source the reviewer cited. For 2018, our base year, the value is 25,562 TWh. For country-specific results, our sample consists of 129 countries (see methods), for which total electricity production in 2018 was 24,805 TWh. Hence, the estimated RE100 demand (227 TWh) equals <1% of the electricity generation in our sample. Please note that these 129 jurisdictions represent almost the entire electricity generation in the world (see response to comment 2 from R2).
P3, l68. How can it already be higher in 2020 (which is the past) than current membership mentioned in line 64?	We thank the reviewer for the detailed scrutiny. We would like to apologize for this typo, which is a follow-on mistake from the one above. The 2018 estimation we mention should be <1% instead of <0.01% (cf. response to previous

	comment). This has been corrected. In the revision, we realized that the write up could be clearer in specifying the exact comparisons when mentioning percentages. We first contrast total electricity demand of RE100 companies with electricity generation in their countries of operation. This gives an indication of the theoretical impact if all RE100 companies switched to 100% RE. This is what the <1% refers to, which is now clearly explained in the revised manuscript (17, 69, 95). Additionally, we created Supplementary Table SI.1, which provides clarity on the referenced percentage numbers. Finally, please note that due to data availability, our values for 2020 are estimations and in Figure 8 (Figure 7 in the original manuscript), we estimate that RE100 companies procured 1.6% of RE generation in the countries of operation, up from 1.4% in 2018 (empirical value). This is now also clearly stated in the revised manuscript (297).
P4, I80, I would say large instead of huge	This has been revised as suggested.
P5, I95, a more detailed comparison based on the supplementary information from (13) would be interesting	We thank the reviewer for this suggestion, which we have carefully considered. We opt against a comparison for two main reasons. First, Lui et al. (ref. 13 in the original manuscript) focuses on comparing the impacts of different international cooperative initiatives therefore analysing the impacts with less granularity. For example, for RE100, the authors do not quantify country-level effects, which is the core of our contribution (see footnotes to Figure 5 and Table 2 in Lui et al.). Follow up reports, such as Kuramochi et al., 2021 provide estimates for ten countries, but do not show the effect of RE100 separately and lack the methodological rigour and the transparency to assure an adequate comparison. Second, Lui et al. assume an expansion scenario of RE100 reaching 1000 members in 2020 and 3000 members in 2030, which we deem too optimistic given current RE100 membership and past growth rates (see methods). Given these methodological differences and the different assumptions, we believe the results are hardly comparable and decided to refrain from a comparison. We hope the reviewer understands our decision and underlying arguments.

P7, l143, what do you mean with 'direct'? Please clarify (in the text)	We thank the reviewer for this remark. We have revised the part to “more likely additional and transformative” and have added two explanatory sentences with an example afterwards (147-151).
P9, l180. The difference between linear and stepwise is an interesting analysis. It would even appeal more to the imagination if you could link it to a understandable number. How much is GWh for example in CO2 terms and/or percentage of 1.5C or 2C budget (400-1000 GtCO2). Or something else	We thank the reviewer for this very interesting suggestion. We have added a comparison to the 1.5C budget and a comparison to fossil fuel emissions of countries for illustration (313-315).
P10, l204-208. What data/projections did you use to determine national 2030 RE shares? This is important to add to the text. Also add this to the caption of Figure 6	We thank the reviewer for this comment. We use the stated policies scenario as per the 2021 World Energy Outlook from the International Energy Agency (see methods). This has been added to the caption of Figure 6 with a reference to the methods.
P11, l226-227, this is worded too strongly (.e.g. 'realistic'). See for example Figure 6.30 in the AR6 WGIII report. Even for a 1.5C scenario, the range of %-low carbon energy is between 70-85%	We thank the reviewer for this remark. Reaching 100% RE in electricity systems around the world is much more challenging than for a selected set of companies to reach 100% RE, namely because of grid infrastructure requirements. We have toned down the sentence referring to the adequacy of such a scenario if RE100 companies wish to be ambitious actors and we believe that reaching 100% RE by 2030 is certainly possible for RE100 companies given the favourable economics around RE.
P17, l343-350 Why did you use all energy data, if electricity only is also available in the CDP dataset, and this is what you are analysing in your research?	We thank the reviewer for this remark. Companies unfortunately do not disclose country-level electricity demand via CDP. Instead, we are able to obtain total electricity demand (TE). To allocate this demand to countries of operation, we are constrained to using the proxy of energy demand, which is often reported at a country-level. We have added a short explainer on this in the methods (420-421) where the process is described in detail.
P19, l402, this formula seems wrong. $TE_{c,2030}$ is both on the left side and right side, so can be crossed out against each other. Then the second term on the right is equal to zero. First term on the right side should probably be $TE_{c,2018}$	We thank the reviewer for the detailed scrutiny. As the reviewer states, $TE_{c,2030}$ should only appear on the left side of the equation. On the right side, the variable is indeed $TE_{c,2018}$. This has been corrected.

Reviewer 2

Comment	Response
This manuscript aims to answer an important policy question of whether corporate climate action initiatives such as RE100 would really deliver substantive mitigation contributions. This research work is timely, as there has been an increasing scepticism against corporate climate pledges following the COP26. I find the quantitative analysis solid overall, but some additional work would be required before the manuscript could be accepted for publication. Below are a few high-level comments, followed by more detailed comments.	We thank the reviewer for the positive overall feedback and the suggestions for improvement, which have helped sharpening the paper substantially. We respond to all comments in detail below. All changes are highlighted in yellow in the revised manuscript and we refer to line numbers in brackets.
1. Readability: Due to the objective of RE100, the text goes back and forth between total electricity generation/demand and (total) renewable electricity generation/demand. This makes it difficult to read through the manuscript even for a person like me, who's more familiar with the topic than most other readers. I do not have a good suggestion, but perhaps good to have the revised manuscript checked by a scientific copyeditor (in case you haven't done it yet).	We thank the reviewer for this comment. The manuscript has gone through a professional editing and proofreading service. Nonetheless, we see the need for more clarity on key terms. First, for the distinction between energy and electricity, we have made sure to consistently use the term electricity when referring to the procurement of RE by companies. In addition to a few instances in the main text, we have replaced energy with electricity in the title, which we think is more precise. Conversely, we consistently refer to the energy transition as this is the term commonly used in academic and policy debates. Second, we have added a sentence clarifying the use of the term "total electricity" demand in the case of companies and generation in the case of countries (lines 49-51). We have also revised the manuscript in a few instances where we referred to electricity production. We now only use the word generation, which is more accurate. Finally, we added Supplementary Table SI.1 to have a clear reference to the percentage value we refer to in the manuscript. We hope these changes help the reader follow the text.
2. Key messages: In the Discussion section, you have the conclusion: "Generally, our study dampens the hope that corporate RE initiatives will have a substantial impact on accelerating the energy transition." To me this is the headline of your work, but the abstract is rather modest, only saying "raising doubts...". I also find a couple of sentences in the abstract to be potentially misleading or not	We thank the reviewer for this very concrete and helpful feedback on the abstract. We now use the wording from the discussion in the revised abstract.

worded right:	
2.1. “We found that these companies source electricity in 129 jurisdictions and account for <0.01% of global electricity production” >> First, if it’s about 129 jurisdictions, then wouldn’t it be better to use the total electricity generation in the 129 jurisdictions as the denominator? Second, it seems that the <0.01% value is a close value of 0.009% in Line 93, but I’m not sure if the figure is correct (see under “Detailed comments” below.	Indeed, there was a typo in the headline number as noted by R1 too. We apologize for this; the numbers are now correct (<1%). We also carefully re-checked all numbers in the manuscript to make sure they are correct. The text was somewhat unclear using the word “global”. We have clarified this and now stipulate clearly that we compare the RE generation to total electricity generation and RE generation in the 129 jurisdictions in the denominator as suggested by the reviewer. These 129 jurisdictions closely represent the global situation because they account for 96% of global electricity generation and for 97% of global RE generation. However, we now abstain from referring to global comparisons in the abstract to be more precise and we have included a sentence in the introduction (70-71) to clarify.
2.2. “By 2030, stringent and frequent interim targets could lead to an additional 391 TWh of RE procured by RE100 companies” >> I understand this to be cumulative additional production between 2020 and 2030. If so, then it should be clarified as readers may mistake it for annual production in 2030.	This has been revised accordingly in the abstract.
3. Analysis: A policy report published last year also attempt to quantify the impact of RE100. Perhaps good to compare the overall results? https://newclimate.org/2021/06/23/global-climate-action-from-cities-regions-and-businesses-2021/	We thank the reviewer for pointing out this New Climate report. We understand it to be the continuation of the authors’ work published in Climate Policy (Lui et al., 2020), which we refer to extensively. There are two challenges in comparing our results to the referenced report. First, the authors of the report state that they calculate RE100 companies’ power and heat generation per geography to obtain country-level data. However, it is unclear whether geography refers to regions (available in the CDP data, see methods) or indeed countries. The authors refer to “confidential CDP data” (see footnote 2 in Annex II) without further explanation. Second, the report does not publish country-level data but only reports global data per initiative (see Table A1) and data for all initiatives together for ten countries (see section 3, Annex I). The main contribution of our paper is the calculation

	of country-level RE100 companies' electricity (and RE) demand, which we deem crucial to assess the impact of RE100 on the energy transition. We emphasize this more strongly in the revised manuscript by expanding on the national policy discussion with the new Figure 7 and by explicitly listing the contributions in the discussion (319-325). Moreover, the report defines additionality as the demand by RE100 companies that is above the average increase in the RE share in OECD countries. While this may work for a global assessment, it does not consider country differences. We prefer comparing different scenarios for the development of RE100 against each other, rather than establishing an overall baseline, which seems hard to defend. However, we acknowledge the need to put our results in context and compare the overall numbers to the ones found in the report in the revised manuscript with a note of caution because of methodological differences (312-313). They align remarkably well. If we use coal-fired power plant emission factors as an upper bound and gas-fired power plant emission factors as a lower bound (both European) as suggested by Lui et al., 2020, we find a range of 40.8 – 97.1 Mt CO₂/y for baseline, 53.7 – 127.9 Mt CO₂/y for more ambitious targets and 349.2 – 831.3 Mt CO₂/y for membership expansion. We discuss the case of gas in the manuscript (310-312). The mentioned report (cited in our paper as Kuramochi et al., 2021) finds a range of 46 – 120 Mt CO₂/y for existing members and 164 – 336 Mt CO₂/y for membership expansion.
4. Analysis: I would like to see more in-depth analysis on the RE sourcing strategies presented in Figure 4. Some strategies contribute more to additional RE generation, while others are simply displacing fossil fuel-fired power generation to other consumers. Deeper qualitative insights into the RE sourcing strategies further strengthen your overall assessment that RE initiatives will have limited impact on accelerating the energy transition.	We thank the reviewer for this comment. Unfortunately, data availability does not allow us to implement a country-level analysis of sourcing strategies, which would indeed be very useful. We address this comment with three modifications. First, we expanded the discussion of sourcing strategies with a focus on their impact or additionality (167-182). Second, we added a new figure to discuss the impact that RE100 commitments may have on country policy (Figure 7). While this is

	not directly linked to sourcing strategies due to the data limitations, it offers a first perspective on where the impact of RE100 could likely materialize. Finally, we mention qualitative research on company-level sourcing strategies as an avenue for future research (361-363).
Detailed comments 1. L35: food companies can be carbon-intensive, depending on the emission scope you look at. If you consider the entire value chain including upstream land use-related emissions, then some food companies are one of the worst (see e.g. “Emissions Impossible” report by IATP).	We thank the reviewer for this remark, with which we agree. We have removed the reference to the food industry because indeed it can be very carbon-intensive depending on the boundaries of the analysis in terms of the value chain that is considered.
2. LLL71: It should be clarified that the 2021 figure is the annual total.	This sentence has been deleted as it was not very clear, and it is mentioned elsewhere in the manuscript.
3. L93: I understood the figure to be global. If so, then the figure might be an order of magnitude too small? Global electricity generation is about 27,000 TWh in 2019-20, so 221TWh would be a bit less than 1%. Or have I missed anything?	We thank the reviewer for the attention. This was a mistake like in the abstract and it should be 0.9% as the reviewer correctly remarks. It has been corrected in the revised manuscript. We have checked all numbers carefully.
4. Figure 5: A cumulative difference of 360,900 GWh is presented, but does this correspond to the 391 TWh figure presented in the abstract and in L70? If yes, please explain the difference. (The Discussion section gives 361 TWh)	We would like to thank the reviewer for the detailed scrutiny. Indeed, the cumulative difference is 360,900 GWh or 361 TWh. The typo has been corrected. None of the comparisons or implications change. We have checked all numbers carefully.
5. Figure 7: The Y-axis should be TWh, not TwH.	This has been corrected.
6. L299: “with interim targets of 60% by 2030 and 90% by 2040” >> I could not find these numbers in the cited report (RE100 annual report 2019: https://www.there100.org/sites/re100/files/2020-09/RE100ProgressandInsightsAnnualReport2019.pdf , and their two annexes). Please provide the page no. where they’re stated.	We thank the reviewer for this observation. The interim targets stem from the RE100 joining criteria. We referenced the annual report as the framework document because the joining criteria do not have a publishing year (see here for the most recent version of Oct 2022, interim targets in Art. 2). We now reference the joining criteria with a reference to the access date (Dec 2019) for accuracy. Interim targets requirements remain unchanged in the most recent version referenced above.
7. L394: Stated Policies Scenario	Corrected.

8. L395: WEO2021 has the historical data for 2019, so why 2018?	We thank the reviewer for this remark. We prefer starting with data from 2018 because the data on RE100 companies is from 2018 too. As such, we ensure a consistent starting year. Using 2019 data (or later) would use information that is not available in our RE100 dataset on other variables, which we think would be less consistent.
9. L402: The first term in the right side of the equation should be Tec,2018?	We thank the reviewer for the detailed scrutiny. Indeed, this was also pointed out by R1 and has been corrected.
10. LL436-438: “We assumed that additional member companies reflect the average size, the average geographical activity distribution and the average RE100 target (level of ambition) of current RE100 members” >> (Clarification) Does this mean that the stepwise trajectory in Figure 5 will be scaled up by about factor 5?	We thank the reviewer for this comment. In the comparison of linear versus stepwise target achievement in Figure 5, we use the baseline scenario only. The purpose being to compare the effects of different temporal patterns rather than mixing it with other scenarios, such as membership expansion. However, the reviewer is of course correct that the difference between the two trajectories (linear and stepwise) would increase if combined with membership growth. It would increase less than five-fold because increase membership from the 185 companies in our sample to 1,000 but we assume a linear growth of membership (see methods). We prefer keeping the two analyses separate to show the effects of interim targets and membership expansion separately.
11. Table M.1: STAPS >> STEPS	Corrected.

REVIEWERS' COMMENTS

Reviewer #1 (Remarks to the Author):

The comments have been addressed sufficiently, and I have only two minor comments on part that need to be cleared in the text to make it fully understandable

1) P7, l147, you say the impact of RE procured is more likely to be additional. But additional to what? That is very important to specify. If it would be 100% additional, that would be very ambitious, as a country already has a X% RE electricity target. I would say, the maximum additionality to national ambitions is (100-X). See, Hsu et al (2019) for different forms of additionality. It would be good to explicitly mention what additionality you are talking about

2) P19, l214-217. It is unclear how you use the country (RE) projections in the trajectories for company RE target achievement. At least, that is what you seem to imply here. A RE target pathway would only consist of interpolating between the current year (2018) and the target, which would not involve any country trends. Why would you need the country growth (you call it 'general RE increase over time')? This is also not described in the Methods section, at least not in 'Defining RE100 progression path towards targets'. It is essential that you describe what you did here. In one or two sentences in the main text, and with possibly more in the Methods.

Reviewer #2 (Remarks to the Author):

Dear authors,

Thank you for the detailed responses to each comment I made on the first manuscript.

All my comments, questions and suggestions are well addressed. I've also gone through the authors' responses to the comments by reviewer #1 and they are also well addressed.

I don't have further comments on the manuscript, and recommend the manuscript to be published.

Reviewer 1

Comment	Response
The comments have been addressed sufficiently, and I have only two minor comments on part that need to be cleared in the text to make it fully understandable.	We thank the reviewer for the time dedicated to our manuscript. We address the remaining two minor comments in detail below.
1) P7, l147, you say the impact of RE procured is more likely to be additional. But additional to what? That is very important to specify. If it would be 100% additional, that would be very ambitious, as a country already has a X% RE electricity target. I would say, the maximum additionality to national ambitions is (100-X). See, Hsu et al (2019) for different forms of additionality. It would be good to explicitly mention what additionality you are talking about	We thank the reviewer for this comment that helped us to be even more precise concerning the concept of additionality in the final version now. We have clarified the statement using the definitions proposed in Hsu et al. (2019). In lines 150-157, we explain that we understand these RE procurements as additional that would not have happened under current policy (i.e., second point in Hsu et al.). RE for self-consumptions, e.g., co-located with a production plant, is an example that we mention in the revised manuscript. We also clarify that we expect a ranking of the sourcing strategies in terms of additionality, while all except for self-consumption are unlikely to be 100% additionally as pointed out by the reviewer.
2) P19, l214-217. It is unclear how you use the country (RE) projections in the trajectories for company RE target achievement. At least, that is what you seem to imply here. A RE target pathway would only consist of interpolating between the current year (2018) and the target, which would not involve any country trends. Why would you need the country growth (you call it 'general RE increase over time')? This is also not described in the Methods section, at least not in 'Defining RE100 progression path towards targets'. It is essential that you describe what you did here. In one or two sentences in the main text, and with possibly more in the Methods.	We thank the reviewer for this note. We clarified the sentence as follows (line 223): "Note that both trajectories show an increase of RE demand in every year because companies' absolute electricity demand is expected to grow in line with electricity growth rates in the countries, they have business operations in (assuming the Stated Policies Scenario by the IEA, see Methods). Hence, even if the RE share remains constant, the absolute RE demand increases slightly." We also added an additional sentence in the Methods: "We assume that companies' total electricity demand grows in line with electricity demand growth in the countries they operate in (see two previous sections)." As the previous methods section describes how electricity demand projections by country are calculated and how companies' electricity demand is allocated to countries, we believe the methods are now comprehensive.

Reviewer 2

Comment	Response
Dear authors, Thank you for the detailed responses to each comment I made on the first manuscript. All my comments, questions and suggestions are well addressed. I've also gone through the authors' responses to the comments by reviewer #1 and they are also well addressed. I don't have further comments on the manuscript, and recommend the manuscript to be published.	We thank the reviewer for the time dedicated to our manuscript and are glad to read the recommendation for publication.